# Pan-Cancer Analysis Reveals Disulfidoptosis-Associated Genes as Promising Immunotherapeutic Targets: Insights Gained from Bulk Omics and Single-Cell Sequencing Validation

**DOI:** 10.3390/biomedicines12020267

**Published:** 2024-01-24

**Authors:** Borui Xu, Minghao Li, Nuoqing Weng, Chuzhou Zhou, Yinghui Chen, Jinhuan Wei, Liangmin Fu

**Affiliations:** 1Department of Pancreato-Biliary Surgery, The First Affiliated Hospital, Sun Yat-sen University, Guangzhou 510080, China; xubr@mail2.sysu.edu.cn; 2Department of Urology, The First Affiliated Hospital, Sun Yat-sen University, Guangzhou 510080, China; limh56@mail2.sysu.edu.cn; 3Department of Gastrointestinal Surgery, The Eighth Affiliated Hospital, Sun Yat-sen University, Shenzhen 518033, China; wengnq3@mail.sysu.edu.cn (N.W.); zhouchzh6@mail2.sysu.edu.cn (C.Z.); 4Department of Intensive Care Unit, Jiangmen Central Hospital, Jiangmen 529030, China; chenyinghui@jmszxyy.com.cn; 5Institute of Precision Medicine, The First Affiliated Hospital, Sun Yat-sen University, Guangzhou 510080, China

**Keywords:** disulfidoptosis, pan cancer, hepatocellular carcinoma, single-cell analysis, immunity infiltration, targeted therapy

## Abstract

Disulfidoptosis, a novel form of cell death, is distinct from other well-known cell death mechanisms. Consequently, a profound investigation into disulfidoptosis elucidates the fundamental mechanisms underlying tumorigenesis, presenting promising avenues for therapeutic intervention. Comprehensive analysis of disulfidoptosis-associated gene (DRG) expression in pan cancer utilized TCGA, GEO, and ICGC datasets, including survival and Cox-regression analyses for prognostic evaluation. We analyzed the association between DRG expression and both immune cell infiltration and immune-related gene expression using the ESTIMATE and TISDIB datasets. We obtained our single-cell RNA sequencing (scRNA-seq) data from the GEO repository. Subsequently, we assessed disulfidoptosis activity in various cell types. Evaluation of immune cell infiltration and biological functions was analyzed via single-sample gene set enrichment (ssGSEA) and gene set variation analysis (GSVA). For in vitro validation experiments, the results from real-time PCR (RT-qPCR) and Western blot were used to explore the expression of SLC7A11 in hepatocellular carcinoma (HCC) tissues and different cancer cell lines, while siRNA-mediated SLC7A11 knockdown effects on HCC cell proliferation and migration were examined. Expression levels of DRGs, especially SLC7A11, were significantly elevated in tumor samples compared to normal samples, which was associated with poorer outcomes. Except for SLC7A11, DRGs consistently exhibited high CNV and SNV rates, particularly in HCC. In various tumors, DRGs were negatively associated with DNA promoter methylation. TME analyses further illustrated a negative correlation of DRG expression with ImmuneScore and StromalScore and a positive correlation with tumor purity. Our analysis unveiled diverse cellular subgroups within HCC, particularly focusing on Treg cell populations, providing insights into the intricate interplay of immune activation and suppression within the tumor microenvironment (TME). These findings were further validated through RT-qPCR, Western blot analyses, and immunohistochemical analyses. Additionally, the knockdown of SLC7A11 induced a suppression of proliferation and migration in HCC cell lines. In conclusion, our comprehensive pan-cancer analysis research has demonstrated the significant prognostic and immunological role of disulfidoptosis across a spectrum of tumors, notably HCC, and identified SLC7A11 as a promising therapeutic target.

## 1. Introduction

In addressing cancer, a paramount health threat, it’s essential to comprehend the mechanisms of cell death [1]. A recent paper has identified a novel cell death pathway known as “disulfidptosis”, which is divergent from other conventional mechanisms such as apoptosis, necroptosis, and oncosis. For cancer cells experiencing glucose starvation, elevated expression of cystine transporter solute carrier family 7 member 11 (SLC7A11; also known as xCT), a cystine transporter, leads to NADPH depletion and abnormal disulfide bonding, which ultimately results in actin network collapse and subsequent cell death [2]. A visual representation of the specific mechanism can be found in Appendix A. As a tumor metabolism-related pathway, disulfide bond formation is found in cancer-related proteins, meaning this metastable bond can be a target for future therapies [3]. In addition, disulfidptosis has been demonstrated to be associated with immune response and prognosis in certain cancers, such as hepatocellular carcinoma (HCC), esophageal squamous cell carcinoma (ESCC), and bladder cancer (BCa) [4,5,6]. Prior research established disulfidptosis as a predictive biomarker for immune characteristics and drug responses, suggesting SLC7A11-mediated disulfidptosis as a novel therapeutic strategy in cancer.

The current consensus is that chronic inflammation, characterized by immune cell infiltration, can promote the initiation or progression of malignancies, including colorectal tumors, renal cell carcinoma, and HCC [7]. Recent evidence reveals that certain cell death modalities can intensify inflammation rather than just result from it. These processes vary in their capacity to provoke inflammation, influenced by the differential release of mediators, affecting dead cell clearance. Subsequent increases in cytokines, chemokines, and reactive oxygen species escalate the inflammatory response [8]. Additionally, targeted therapy induces significant modifications in the transcriptional dynamics within cancer cells and the adjacent tumor milieu, which are pivotal in determining the emergent tumor phenotypes associated with pharmacological sensitivity or resistance [9]. In the treatment of HCC, traditional chemotherapy agents like doxorubicin and fluorouracil have shown limited efficacy and are associated with substantial adverse effects [10]. Recent studies have witnessed groundbreaking randomized controlled trials in advanced hepatocellular carcinoma, heralding significant shifts in clinical practice [11]. These include the introduction of lenvatinib as a first-line alternative to sorafenib and the second-line use of regorafenib, cabozantinib, and ramucirumab following sorafenib. Among them, regorafenib and cabozantinib were suggested to be chosen as second-line therapies [12]. Moreover, the efficacy of immunotherapy has been cemented in HCC treatment through the combination of the anti-PD-L1 agent atezolizumab and the anti-vascular endothelial growth factor (VEGF) agent bevacizumab, which gradually becomes a first-line choice [13]. Prior studies show that SCL7A11-mediated ferroptosis in tumor-associated macrophages significantly alters the hepatocellular carcinoma microenvironment. This underscores the potential of targeting SCL7A11 in HCC treatment, suggesting that combining SCL7A11 with immunotherapy could enhance therapeutic efficacy clinically [14]. The mechanisms by which SLC7A11-mediated disulfideptosis operates and influences the immune microenvironment and inflammation remain nascent. Consequently, additional research is needed to clarify disulfideptosis’s role within the tumor microenvironment.

Given the significant role of disulfidoptosis in cancer, we analyzed pan-cancer data across multiple databases. We aimed to investigate the differential expression, prognostic significance, and biological implications of disulfidoptosis-related genes (DRGs) in various tumors. Critical insights from single-cell sequencing further illuminated the intricate cellular heterogeneity of HCC, often hidden in traditional bulk RNA sequencing approaches, emphasizing the nuanced roles that DRGs play within this cellular mosaic [15]. We observed different immune statuses and biological functions in HCC with differential DRG expression, underlining the pivotal role disulfidptosis has in shaping TME characteristics in HCC. The expression of SCL7A11 was confirmed at both mRNA and protein levels using qRT-PCR and Western blotting in clinical samples and cell lines. And the knockdown of SLC7A11 resulted in diminished proliferation and migration in HCC cell lines. Our research methodology is delineated in Figure 1. Our findings highlight disulfidoptosis as a promising biomarker and therapeutic avenue for malignancies, notably hepatocellular carcinoma.

## 2. Materials and Methods

### 2.1. Data Collection and Processing

According to the study of Xiaoguang Liu, a total of 10 genes were identified as key genes most associated with disulfidoptosis, which were annotated as DRGs [2]. The genomic, clinicopathological, and somatic mutation data of 33 TCGA GDC pan cancers were gathered from the University of California Santa Cruz (UCSC) Xena (https://xenabrowser.net/, accessed on 6 June 2023). The fragment per kilobase million (FPKM) value was normalized before comparison. The abbreviations of various cancers are provided in Appendix A.

The GSE242889 dataset encompassed single-cell RNA sequencing details of five HCC specimens based on the 10X Genomics platform. The dataset was constructed on the platform GPL24676 via the Illumina NovaSeq 6000 system. To meticulously scrutinize the scRNA-seq dataset, a stepwise analytical framework was invoked. As a starting point, data preprocessing was accomplished through the Seurat toolkit. This stage paved the way for a comprehensive series of analyses. Utilizing the PercentageFeatureSet function, we delineated the proportion of mitochondrial gene presence. Additionally, we performed correlational evaluations to probe relationships among sequencing depth, mitochondrial gene content, and overall intracellular transcriptomes.

To bolster the precision of our analysis, only genes expressed in a minimum of five cells were considered. Cells were stringently curated based on explicit benchmarks: gene expression counts lying between 300 and 5000, a mitochondrial constitution below 10%, and an essential UMI threshold of 1000 per cell. This rigorous curation was geared towards ensuring cellular data fidelity. After the data filtering process, we normalized the scRNA-seq data using the LogNormalize method, thereby enhancing the accuracy of downstream analysis and interpretation.

For downstream analysis, the top 20 principal components (PCs) were subjected to Seurat’s Elbow plot program. With a resolution of 0.7, primary cell clusters were identified by Seurat’s Find Clusters tool, which were subsequently visualized via two-dimensional UMAP plots. Cells were categorized into previously recognized biological cell types using conventional markers established in former studies.

### 2.2. Survival Analysis

Based on the median expression levels of DRGs, patients were stratified into low- and high-expression cohorts. We employed univariate Cox regression analysis on TCGA datasets, utilizing the R packages “survival” and “forestplot”, to assess the prognostic significance of DRGs regarding overall survival (OS). The derived log-rank *p*-values and hazard ratios (HRs), complemented with 95% confidence intervals, are illustrated in forest plots to provide a comprehensive representation of the survival analysis. Additionally, KM survival analysis was executed to compare OS between the two defined expression cohorts, facilitated by the “survminer” and “survival” packages in R.

### 2.3. Immune Subtypes and Stemness Score Analysis

The immune subtype analysis of pan cancer was performed utilizing the “limma”, “ggplot2”, and “reshape2” packages in R. The *p*-value was set at 0.05 as a threshold for statistical significance.

### 2.4. DNA Promoter Methylation Analysis

The GSCA database (http://bioinfo.life.hust.edu.cn/GSCA, accessed on 6 June 2023) was employed to scrutinize the promoter methylation distribution of DRGs and the correlation between DNA promoter methylation levels and DRG mRNA expression, as well as survival outcomes in diverse cancers. Correlations are depicted through detailed scatter diagrams, and Spearman’s correlation coefficient was applied to calculate correlation values (Cor) and false discovery rates (FDR), offering a robust quantitative analysis of the interdependencies.

### 2.5. Immune Infiltration Analysis

The ESTIMATE algorithm, which is a computational method designed to infer the fraction of immune and stromal cells within tumor samples by analyzing signatures of gene expression, has been used to calculate the ImmuneScore (Immune Component Ratio), StromalScore (Matrix Component Ratio), and ESTIMATEScore (Sum of ImmuneScore and StromalScore) for tumor purity prediction [16]. Spearman correlation was used to calculate the correlation between these scores and DRG expression, and the results are presented as scatter plots with the *p*-value and Cor.

TME cell infiltration was evaluated using a single-sample gene set enrichment analysis. Utilizing the “GSVA” R package, we conducted a single-sample gene set enrichment analysis (ssGSEA) to gauge the infiltration rates of various immune cell populations. The ssGSEA technique interrogates individual oncological specimens based on gene profiles characteristic of specific immune cells [17]. We conducted correlation analysis using the Spearman method to assess the relationship between essential genes and 24 immune cell types. Adopting a deconvolution methodology, we assessed the presence of 24 distinct immune cell types, which encompassed activated B cells, CD4+, and CD8+ T cells in their activated states, dendritic cells (both activated and immature forms), natural killer cells characterized by CD56 bright and CD56 dim phenotypes, eosinophils, gamma delta T cells, immature B cells, myeloid-derived suppressor cells (MDSCs), macrophages, mast cells, monocytes, NK-T cells, neutrophils, plasmacytoid dendritic cells, regulatory T cells, T follicular helper cells, and the T helper cell subsets—Th1, Th2, and Th17 [18].

### 2.6. Single-Sample Gene Set Enrichment Analysis (ssGSEA) and Gene Set Enrichment Analysis

In biomedical analysis, ssGSEA is essential for calculating gene set enrichment scores within a single sample. The ssGSEA score serves as an indicator, elucidating the extent of systematic upregulation or downregulation of a designated gene set within a sample [19]. We derived DRG scores for each sample using ssGSEA via the R “GSVA” package. We employed the “GSVA” package in R to execute a gene set variation analysis, delving into the biological functionalities and potential pathways linked to the expression of DRGs. Prior to this analysis, the gene set designated as “c2.cp.kegg.v7.5.1.symbols” was extracted.

### 2.7. Cell Culture

Human osteoblast cells (hFOB1.19) and human osteosarcoma cells (U2OS) were gifts from Dr. HJ. Lu, Guangdong Pharmaceutical University. Human renal cancer cells (786-0) and human kidney cells (HK-2) were gifts from the research group of Prof. JH. Luo, Department of Urology, The First Affiliated Hospital of Sun Yat-Sen University. Human hepatocellular carcinoma cells (SNU449 and Huh7) and human normal hepatic cells (LO2) were acquired from the Shanghai Institutes for Biological Sciences, Chinese Academy of Sciences (ATCC, Shanghai, China). Human U2OS and hFOB1.19 cell lines were maintained in DMEM media (Gibco, Thermo Fisher Scientific, Suzhou, China) enriched with 10% fetal bovine serum (ABW, AB-FBS-1050S, Uruguay) and 1% penicillin–streptomycin (Invitrogen, Carlsbad, CA, USA). The 786-0, HK-2, SNU449, Huh7, and LO2 cell lines were cultured in RPMI 1640 media (Gibco, Thermo Fisher Scientific, Suzhou, China) supplemented identically. Cells were cultured at 37 °C in a 5% CO_2_ atmosphere at a constant humidity of 80%. Cells in the logarithmic growth stage, approaching 80% confluence, were selected for subsequent experiments.

### 2.8. Clinical Samples Collection

Samples of hepatocellular carcinomas and adjacent normal tissues were collected from patients admitted to the First Affiliated Hospital of Sun Yat-sen University inbetween 2019 and 2020. The patients range in age from 18 to 80. The patients did not undergo neoadjuvant chemotherapy or radiotherapy prior to surgery. All samples were obtained through surgical resection and were processed and stored in liquid nitrogen containers within half an hour post-surgery. A review and approval of this study were obtained from the Institutional Ethics Committee for Clinical Research of the First Affiliated Hospital of Sun Yat-sen University ((2021)170). We conducted all experimental procedures in accordance with the Helsinki Declaration and obtained written informed consent from all patients.

### 2.9. RNA Isolation and Quantitative Reverse Transcription-Polymerase Chain Reaction (qRT-PCR) Analysis

The results of this study were verified by quantitative real-time PCR (qRT-PCR) after RNA was isolated from 10 different cell lines, including hFOB1.19, U2OS, 786-0, KH-2, SNU449, Huh7, LO2, and si-SLC7A11-transfected SNU449 or Huh7, as well as 8 pairs of HCC tissues and adjacent normal hepatic tissues. As part of the RT-qPCR process, primer sequences for the relevant genes are found in Appendix A and are synthesized by RiboBio (Guangzhou, China). As stated in Appendix A, each primer has its own melting temperature (Tm) and annealing temperature (Tm-5 °C). The total RNA was extracted by the EZ-press RNA Purification Kit (EZBioscience, ZScience Biotechnology Corporation Limited, Roseville, MN, USA), and the purity and concentration of the total RNA were assessed using a NanoDrop 2000 spectrometer. As specified by the manufacturer, the RT-qPCR reaction system and conditions were conducted using the 4×Reverse Transcription Master Mix (EZBioscience, ZScience Biotechnology Corporation Limited, Roseville, MN, USA) and 2×SYBR Green qPCR Master Mix (EZBioscience, ZScience Biotechnology Corporation Limited, Roseville, MN, USA) on the Applied Biosystems™ QuantStudio™ 5 Real-Time PCR System. Drawing the figure was performed using Graphpad Prism 9.0 software (GraphPad Prism 9.0, https://www.graphpad-prism.cn/, accessed on 23 February 2022, China). In order to analyze the RT-qPCR data, we used GAPDH as an internal reference, and the 2-ΔΔCt method was used to normalize the expression of the genes targeted. In the studies, cells were used when the confluence rate reached approximately 80% while they were in the logarithmic growth stage. 

### 2.10. Western Blotting and Immunohistochemical

We lysed cell lines and four pairs of hepatocellular carcinoma tissues using RIPA lysis buffer (Beyotime, Shanghai, China) containing a Protease Inhibitor Cocktail (CoWin Biosciences, Jiangsu, China) (1:100). A bicinchoninic acid (BCA) protein assay kit (Thermo Scientific, Guangzhou, China) was utilized to quantify total protein. Following protein quantification, 10 ug of every protein sample was loaded onto a 10% sodium dodecyl sulfate polyacrylamide gel electrophoresis (SDS-PAGE). A 0.45 μm polyvinylidene difluoride membrane was used to transfer total proteins after electrophoresis. Further blocking of the membranes was conducted in a protein-free rapid blocking buffer (Epizyme Biotech, PS108) for 10–15 min. An ECL chromogenic kit (Thermo Fisher Scientific, Guangzhou, China) was used to detect antibody binding after sequential incubation with primary antibodies (overnight at 4 °C) and secondary antibodies (1 h at room temperature). The imaging systems using chemiluminescence (Beijing, China) and the Fusion FX5 image analyzer (Vilber Lourmat, Marne La Vallée, France) were used to visualize antibody binding. The following primary antibodies were as follows: SLC7A11 (Proteintech, Wuhan, China), GAPDH (CST, NewYork, NY, USA), and β-Actin (CST).

For immunohistochemical (IHC) analysis, sections derived from human hepatocellular carcinoma and matched normal liver tissues were used. IHC staining was performed at Wuhan Servicebio Technology. Anti-CD206 Rabbit pAb (GB113497-100; Servicebio, Wuhan, China), anti-SLC7A11/xCT Rabbit pAb (GB115276-100), and anti-FOXP3 Rabbit pAb (GB112325-100; Servicebio) were used to assess macrophages and Treg cells on paraffin-embedded tumor tissue and matched normal tissue slides. Images were captured with the OCUS portable digital scanning microscopic imaging system (APG Bio, Shanghai, China). Following a quick examination of slide quality, the IHC results were evaluated by two independent pathologists. The intensity of IHC staining and the proportion of positively stained tumor cells were distinctly assessed, as detailed below. The percentage of stained tumor cells was recorded as follows: 0%: 0; <1%: 1; 1–10%: 2; 11–33%: 3; 34–66%: 4; and >66%: 5. The degree of staining intensity was categorized as: 0; weak staining: 1; moderate staining: 2; strong staining: 3. IHC staining score = the percentage of stained tumor cells + staining intensity. These IHC results were grouped according to IHC staining score as: negative (IHC staining score: <3), weakly positive (IHC staining score: 4–6), and strongly positive (IHC score: 7–8).

### 2.11. siRNA Interference Assay and CCK-8 Assay

Two SLC7A11-siRNAs were designed by HanYi Biosciences Inc., Guangzhou, China (Appendix A). Transfecting SNU449 or Huh7 with siRNAs was performed using jetPRIME (Polyplus, Illkirch, French) with the manufacturer’s instructions. A functional assay was conducted 48 h after transfection, and proteins and RNA were harvested.

After transfection with SLC7A11-siRNA, cell proliferation was assessed using the Cell Counting Kit-8 (Dojindo, Japan, CK04-5). A 96-well plate was used to seed the cells. A microplate reader (Bio-Rad Laboratories, Hercules, CA, USA) was used to measure cell viability at 1, 2, 3, 4, and 5 days.

### 2.12. Wound Healing, Cell Migration Assays, Colony Formation Test

A pipette was used to scratch the cells after they had been merged into the six-well plate. After scratching, we took photos at 0 h and 76 h. SNU449 or Huh7 were starved in serum-free RPMI 1640 medium for 8 h in order to evaluate their migration capability. Next, 5 × 10^4^ cells in 100 μL of serum-free RPMI 1640 medium were added to transwell inserts (Corning, NY, USA). As a nutritional attractant, serum-free RPMI 1640 medium with 10% FBS was used as a base for transwell assays. After 8 h, lower surface cells were fixed with 4% polyformaldehyde (Beyotime) for 30 min and stained with 0.4% crystal violet (Beyotime). The upper surface of the slide was wiped clean using a cotton swab, and the lower surface was counted under the microscope. Huh7 or SNU449 transfected with SLC7A11-siRNA were incubated in RPMI-1640 medium containing 10% FBS, maintained in RPMI-1640 medium containing 10% FBS, and incubated at 37 °C with 5% CO_2_. Within 24 h of transfection, we inoculated 1000 Huh7 or SNU449 cells into the six-well plates, which were then cultured in RPMI-1640 medium with 10% FBS for two weeks, and finally we counted and analyzed the colonies.

### 2.13. Statistical Analysis

The Kruskal–Wallis test and Wilcoxon rank-sum analysis were used to evaluate differences in gene expression levels between tumors and normal tissues. We used two-tailed tests and then set a significance threshold of *p* < 0.05 for all parametric analyses in this study. The statistical analyses were conducted using the R software (version 4.1.3, manufacture, city, if any state, country R Foundation for Statistical Computing), which was obtained from the CRAN mirror hosted by Tsinghua University, Beijing, China (https://www.r-project.org/), on 4 March 2023. The Fisher’s exact test and the *t*-test, assuming equal variances, were used to evaluate group comparisons for categorical and continuous variables unless explicitly specified. We evaluated the diagnostic precision of gene expression levels in predicting preeclampsia, and we conducted an ROC curve analysis and calculated the AUC values. Statistical significance was set at *p* < 0.05, unless otherwise noted.

## 3. Results

### 3.1. Pan-Cancer Gene Expression Profiles of Disulfidoptosis-Related Genes

In our study, we delineated the expression profile of DRGs in pan-cancer. All 10 genes were expressed in the tissue, with RPN1 exhibiting the highest expression level and SLC7A11 the lowest (Figure 2A). The comparison of the differential expression of DRGs between tumor and adjacent normal tissues across all TCGA tumors revealed that in multiple types, the DRGs expressed differentially, yet the overexpression or underexpression varied across diverse cancer types. And SLC7A11 demonstrated notably differential expression relative to other genes in the set (Figure 2B). The correlation between them was further confirmed by the relationship map between genes. For instance, based on the results of the correlation analysis, the expression levels of DRGs increased concomitantly with the upregulation of SLC7A11, with the exception of NDUFA11 and OXSM (Figure 2C). For instance, SLC7A11 was overexpressed in liver hepatocellular carcinoma (LIHC) but underexpressed in glioblastoma multiforme (GBM), which indicated that the mechanisms for disulfidoptosis in different cancer types were not the same. Liver cancers (LIHC and Cholangiocarcinoma (CHOL)) had a higher probability of occurring disulfidoptosis compared to adjacent noncancerous tissues because their oxidative stress response genes NDUFS1 and mitochondrial gene NUBPL were downregulated and pro-cuproptosis genes SLC7A11 were upregulated in cancer tissues. Conversely, GBM was found to potentially have a lower disulfidoptosis cell death rate in cancer tissue because their pro-disulfidoptosis gene SLC7A11 is downregulated in cancer tissues. Additionally, utilizing the TCGA database (https://portal.gdc.cancer.gov/projects/TCGA-LIHC, access date: 6 June 2023), we examined the distribution and differential expression of genes within the disulfidoptosis gene set across various tumor tissues. As depicted in Appendix A, despite several cancer types either lacking or having limited normal tissues as controls, DRGs in the majority of the remaining cancer types exhibited significant expression differences. Notably, both SLC7A11 and SLC3A1 demonstrated marked differential expression across almost all cancer types. These data strongly suggest that disulfidoptosis may serve as a promising biomarker and tumor-therapeutic strategy, particularly in hepatocellular carcinoma, which exhibits elevated disulfidoptosis-related cell death activity.

Given the pivotal role of SLC7A11 in disulfidoptosis mechanisms and its significant differential expression, as highlighted above, we sought to validate the disulfidoptosis-related cell death activity in these tumor types by assessing the expression levels of SLC7A11 in three cancer cell lines and their normal counterparts using qRT-PCR and Western blot analyses. Then, qRT-PCR was subsequently employed to compare the mRNA transcript levels of SLC7A11 in three cancer cell lines and their normal counterparts, including the 786-0 cell line, the HK-2 cell line, the SNU449 cell line, the LO2 cell line, the U20S cell line, and the hFOB1.19 cell line. The qRT-PCR results demonstrated a significant upregulation of SLC7A11 expression in the cancer cell lines compared to their corresponding normal counterparts (Figure 2D). The significant upregulation of mRNA found in these cell lines was further validated at the protein level in Western blot analysis (Figure 2E).

### 3.2. Prognostic Value of Disulfidoptosis-Related Genes in Pan Cancer

To further explore the prognostic value of DRGs in pan cancer, a risk forest plot and Kaplan–Meier analysis were used to explore the relationship between DRG expression level and survival outcomes. In the survival analysis, identical DRGs exhibited differential prognostic implications across various cancer types (Figure 3A). Specifically, SLC7A11 demonstrated a positive correlation with OS in ovarian serous cystadenocarcinoma (OV), while showing an inverse correlation with OS in LIHC, adrenocortical carcinoma (ACC), kidney renal papillary cell carcinoma (KIRP), mesothelioma (MESO), sarcoma (SARC), thyroid carcinoma (THCA), and uveal melanoma (UVM) (Figure 3B). The Kaplan–Meier analysis results for other DRGs can be found in Appendix A. The results from the aforementioned forest plot indicate that nearly all DRGs are associated with the survival outcomes of patients with LIHC. The findings from the Kaplan–Meier analysis further corroborated this observation. Taking into account the results of both the Cox analysis and the KM survival analysis, our further analysis of the DRGs was focused on LIHC.

### 3.3. Methylation of Disulfidoptosis-Related Genes in Cancers

Methylation could potentially influence the aberrant expression of these genes [18]. Consequently, based on the UCSC database, the differential methylation levels of the DRGs between cancerous and normal tissues were investigated. The top three distinct cancer types were LIHC, lung squamous cell carcinoma (LUSC), and lung adenocarcinoma (LUAD) (Appendix A). The correlation between methylation and gene expression indicated that most of the DRGs demonstrated an inverse relationship with methylation. For instance, in most cancer types, the expression of SLC7A11, OXSM, and NDUFS1 was negatively correlated with methylation. Meanwhile, the expression of LRPPRC and NDUFA11 showed no significant association with methylation (Appendix A). Moreover, the survival correlation analysis of methylation indicated that, in the majority of cancer types, methylation levels are not linked to patient survival across various cancers, including DFI, DSS, OS, and DFS. However, an exception observed is that SLC7A11 methylation levels tended to correlate with worse cancer patient survival, especially in LIHC (Appendix A). These findings highlight the crucial role of SLC7A11 in the mechanism of disulfidoptosis. By regulating the methylation status of SLC7A11, there exists an opportunity to modulate disulfidoptosis activity in tumors, thereby aiming to improve the outcomes of cancer patients.

### 3.4. Correlation of Disulfidoptosis-Related Genes Expression with Tumor Immune Microenvironment

We identified six immune subtypes spanning cancer tissue types and molecular subtypes to define immune response patterns having an influence on prognosis, including C1 (wound healing), C2 (INF-γ dominant), C3 (inflammatory), C4 (lymphocyte depleted), C5 (immunologically quiet), and C6 (TGFβ dominant). Among these pattens, C3 and C5 immune subtypes were associated with a better prognosis, while types C4 and C6 had survival disadvantages [20]. Subsequently, an exploration was conducted to determine if the expression of DRGs correlates with immune subtypes across pan cancer, offering potential insights into the prognosis of cancer patients. As shown in Figure 4A, in the pan-cancer data (all *p* < 0.001), the levels of 10 DRGs were differentially expressed across different immune subtypes. Additionally, the relationship between DRGs and ImmuneScore, Stromalcore, and tumor purity in pan cancer was explored. The ESTIMATE algorithm was applied to determine the relationship between the estimated immune/stromal scores and the evaluated DRGs. TME analysis showed that DRGs expression was negatively correlated with ImmuneScore and Stromalcore while positively correlated with tumor purity in pancancer (Figure 4B–D). These results suggested that disulfidoptosis is linked to changes in the TME and immune infiltrate. Furthermore, they all indicated that patients with high expression of DRGs are likely in an immunologically desert TME state, which could potentially explain the tumorigenic properties of these genes.

### 3.5. scRNA-Seq Analysis of Ten Disulfidoptosis-Related Genes in HCC

Following this, we further investigated the pivotal role of DRGs in the prognosis and immunology of hepatocellular carcinoma. After data filtering, we examined these results, namely the scRNA sequencing dataset, using UMAP techniques. Upon quality analysis of scRNA-Seq data, we excluded zero low-quality cells. Then, we observed a markedly positive relationship between the number of detected DRGs and the total number of genes identified via sequencing. Appendix A demonstrates a strong association between the number of UMIs and mRNAs; however, no evident correlation was observed between the number of UMIs/mRNAs and the content of mitochondrial genes. Violin plots representing pre- and post-quality assurance are shown in Appendix A.

After log-normalization and dimensionality reduction, 30 clusters were identified (Figure 5A). Recognizing the pivotal role that Treg cells play in the immune microenvironment and immunotherapy of HCC, we selected 8 Treg cell marker genes: FOXP3, CCR8, TNFRSF8, LAYN, TNFRSF9, IKZF2, RTKN2, CTLA4, BATF, and IL21R [21]. Within the 30 clusters, these Treg cell marker genes were predominantly expressed in clusters 12, 16, 25, and 28. Using clusters that primarily expressed the eight Treg cell marker genes, further dimensionality reduction clustering analysis was performed (Figure 5B). In measuring disulfidoptosis activity across varied cell types, we employed the “AddModuleScore” function within the Seurat package. This enabled the assessment of expression levels associated with the ten-gene set integral to disulfidoptosis in every cell under study (Figure 5C). Figure 5D displays the UAMP plot representing the distribution of the five samples. Consequently, five distinct Treg cell clusters were identified and utilized for the subsequent analysis.

Based on the DRGs we identified, we observed the expression of DRGs across different clusters. The figure depicts the proportion of the five clusters in each cohort. As depicted in Figure 5E, the KEGG pathway analysis illuminated that the differentially expressed genes (DEGs) were predominantly involved in several key pathways, including the cell cycle, motor proteins, cellular senescence, and DNA replication, to name a few.

### 3.6. Mutation Landscape of Disulfidoptosis-Related Genes in Cancers

To delve into the potential roles of DRGs in tumorigenesis and to discern the underlying causes for their aberrant expression, we undertook an exhaustive pan-cancer analysis. Initially, we probed the CNV patterns of DRGs across 33 distinct cancers in the TCGA database. The heatmap reveals extensive CNV variation for these genes in BLCA, breast invasive carcinoma (BRCA), cervical squamous cell carcinoma and endocervical adenocarcinoma (CESC), head and neck squamous cell carcinoma (HNSC), LUAD, and lung squamous cell carcinoma (LUSC). Notably, RPN1, NUBPL, and OXSM demonstrate widespread CNV across various cancers. Conversely, the expression of SLC7A11 displays a weak negative correlation with CNV and exhibits no correlation with COAD, GBM, or rectum adenocarcinoma (READ) (Figure 6A). To further elucidate the relationship between CNV and gene expression, we classified the samples into CNV-gain and CNV-loss groups. And the result revealed that in the CNV-gain group, the expression levels of the DRGs were elevated compared to samples without CNV, while in the CNV-loss group, the expression levels of the DRGs were diminished compared to those without CNV (Figure 6B). CNV gene frequency analysis showed that NDUFS1, RPN1, NCKAP1, and LRPPRC had a high frequency of amplified copy numbers, whereas NDUFA11 showed a high proportion of reduced copy numbers (Figure 6C). These results indicate that the SLC7A11 gene is less likely to undergo copy number alterations.

Moreover, these risk model genes were examined for single-nucleotide variations (SNVs). The SNV analysis revealed that SLC7A11 and LRPPRC were the two most frequently mutated DRGs. For example, UCEC had 34 LRPPRC SNV samples and 26 SLC7A11 SNV samples among 531 samples, while SKCM had 16 LRPPRC SNV samples and 20 SLC7A11 SNV samples among 468 samples. (Figure 6D) The SNV-landscaped plot of DRGs in cancers clearly depicts the distribution of various SNV types across different cancer types. And LRPPRC and NCKAP1 have high SNV in CESC and SKCM tumors. (Figure 6E) The SNV landscape of 10 DRGs in LIHC is shown in Figure 6F,G, which reveals that TP53 was the most frequently mutated gene, followed by TTN. Missense mutation and C>T ranked the main variant classification and SNV class, respectively. The aforementioned findings suggest that the associated genes undergo fewer SNVs in LIHC, and notably, no SNVs in the *SLC7A11* gene were detected.

### 3.7. Gene Set Enrichment Analysis and Immune Landscape of the DRGs in HCC

Exploring further, potential pathways affiliated with each predisposition gene were evaluated. Figure 7A highlights that 24 unique pathways have significant ties to DRGs, including cell cycle modulation, fatty acid metabolism, and the peroxisome. It is noteworthy that the expression of DRGs is positively correlated with ubiquitin-mediated proteolysis, gap junction, pathways in cancer, regulation of actin cytoskeleton, and DNA replication, while it is negatively correlated with beta-alanine metabolism, drug metabolism, cytochrome P450, histidine metabolism, linoleic acid metabolism, and the PPAR signaling pathway (Figure 7B).

In the oncologic context, the tumor immune microenvironment plays an essential role in mediating tumor initiation, survival, and progression [22]. Integral to this microenvironment, tumor-infiltrating lymphocytes modulate tumorigenesis, invasion, and metastasis by altering the immune status of the neoplastic cells. Furthermore, tumor cells engage in intricate interactions with these immune cells, ultimately compromising their function and facilitating immune evasion [23]. Therefore, we investigated whether the LIHC immune microenvironment was associated with the expression of DRGs. After evaluating the immune microenvironment of LIHC using the ESTIMATE algorithm, we observed a negative correlation between the expression of DRGs and the immune-, stromal-, and ESTIMATE scores, except for the *GYS1* gene (Figure 7C). Analyses unveiled a marked positive association between DRGs and T cells, both CD4-naive as well as regulatory T cells. Additionally, DRGs were robustly linked with M0 macrophages and mast cells resting (Figure 7D). An IHC assessment, showcased in Figure 7E,F, underscored the pronounced presence of macrophages and Tregs within HCC. Given the significant roles played by macrophages and Tregs in tumor evolution, coupled with the results of functional enrichment analysis, it is conceivable that DRGs steer the trajectory of HCC progression.

### 3.8. External Validation of SLC7A11 Expression

To identify hepatocellular carcinomas in which SLC7A11 may be involved in their pathogenesis, the expression level of the SLC7A11 gene was examined in eight pairs of surgically resected HCC specimens and corresponding adjacent normal tissues. Significantly higher expression of SLC7A11 mRNA in hepatocellular carcinoma tissues was confirmed by RT-qPCR analysis (Figure 8A). Consistently, WB analysis was used to measure SLC7A11 expression in hepatocellular carcinoma tissues at the protein level. As shown in Figure 8B, hepatocellular carcinoma tissues (referred to as T) showed higher levels of SLC7A11 protein than paired normal adjacent tissues (referred to as N). Additionally, the result was substantiated through IHC (Figure 8C). Overall, the results clearly validated the possibility that SLC7A11 could be a good target for hepatocellular carcinoma treatment in certain situations.

### 3.9. Effect of SLC7A11 Silencing in LIHC Cell Lines

To explore the role of SLC7A11 in LIHC cell lines, we constructed SLC7A11 knockdown cell lines using transfecting siRNAs. After transfection with siRNA-SLC7A11, we performed an RT-qPCR assay to confirm the efficiency of transfection, and the following result indicated that the expression of SLC7A11 mRNA in SNU449 and Huh7 cell lines was suppressed in the groups transfected with siRNA-SLC7A11-1, -2 compared with the blank control group (each *p* < 0.05; Figure 9A). Next, we assessed SLC7A11 protein expression in the same cell lines. SLC7A11 protein expression was decreased significantly in siRNA-SLC7A11-transfected cell lines, depending on Western blots (Figure 9B). According to these results, siRNA-SLC7A11 reduced the expression of SLC7A11 in four hepatocellular carcinoma cell lines. SNU449 and Huh7 cell lines with si-SLC7A11-1, -2 were then used in subsequent experiments.

### 3.10. SLC7A11 Accelerates Growth, Migration, and Invasion of Hepatocellular Carcinoma Cell Lines

Metastasis is one of the biggest threats affecting the prognosis of cancer patients. The effects of SLC7A11 on malignant biological behaviors of hepatocellular carcinoma cell lines were then examined by CCK8, which include healing, colony formation, and transwell migration assays, for the reason that cell migration and invasion are the initial stages of metastasis. Utilizing CCK-8 assays, we assessed the role of SLC7A11 in the proliferation of hepatocellular carcinoma cell lines, SNU449 and Huh7, post-transfection with si-SLC7A11-1 and -2, and a notable decline in proliferation rates was observed when compared to the blank control over 120 h, initiating 48 h following transfection (*p* < 0.05; Figure 9C). Further, we probed the impact of SLC7A11 silencing on the proliferative ability of these cell lines through wound healing and colony formation assays. Upon scratching confluent SNU449 and Huh7 cell layers, a consistent cell-free gap was discerned. At the 72 h mark, wound healing data demonstrated a diminished migration capacity into the ‘wound’ zone for cells transfected with siRNA-SLC7A11-1 and -2. Compared with the blank control group, siRNA-SLC7A11-1, -2 significantly decreased wound closure percentage (each *p* < 0.05; Figure 9D). The transwell assays manifested a reduction in both migration and invasion capacities of the SNU449 and Huh7 lines post-SLC7A11 silencing (each *p* < 0.05; Figure 9E). Moreover, colony formation results revealed that colony formation ability was greatly impaired under SLC7A11 knockdown (each *p* < 0.05; Figure 9F). Collectively, these findings highlight that SLC7A11 inhibition substantially curtails the proliferation, migration, and invasion capabilities of hepatocellular carcinoma cell lines.

## 4. Discussion

Cell proliferation and programmed cell death always play critical roles in the development of tumors [23]. Previous research has established that most (perhaps even all) types of cancer cells are insensitive to apoptosis; therefore, inducing tumor cell non-programed necrosis is an effective way to treat the tumor [24]. Emerging studies have shown that cuproptosis, ferroptosis, and proptosis play crucial roles in the occurrence and treatment of tumors [25,26]. Given that previous studies have shown that inflammation can augment cell death and increase cellular turnover, promoting tumorigenesis [7]. In this context, apoptosis, characterized as a form of cell death that is devoid of inflammatory responses, emerges as a preferred clinical approach, especially in procedures like tumor ablation [27]. Consequently, a deeper exploration of cell death promises insights into the foundational mechanisms of tumorigenesis, providing avenues for devising more potent antitumor therapeutic strategies.

Disulfidoptosis, a novel form of cell death distinct from necroptosis and apoptosis, highlights the role of disulfide and glucose metabolism in tumor cell death [2]. Extracellular cystine transporter SLC7A11 promotes cysteine synthesis, maintaining intracellular glutathione levels to prevent cell death from abnormal disulfide bonds and reactive oxygen accumulation [25]. ROS accumulation is a common feature in disulfidoptosis, ferroptosis, and cuproptosis, influencing inflammation-related diseases like cancer. ROS have a dual role in tumors, promoting cell death at high levels while stimulating tumor growth [28]. The intricate relationship between disulfidoptosis and energy metabolism warrants further investigation, particularly in HCC, for potential therapeutic insights.

HCC ranks as the sixth most common cancer globally, characterized by a poor prognosis and short survival rates [29]. Sorafenib, having gained approval from the US Food and Drug Administration for HCC treatment, is extensively utilized based on its efficacy in hindering tumor cell proliferation and angiogenesis [30]. However, as the current first-line systemic drug, sorafenib faces potential discontinuation due to its limitations associated with multiple adverse events [31]. And the standard second-line treatments for HCC are scarce. Recent studies have outlined the safety and efficacy profiles of capecitabine, regorafenib, cabozantinib, and ramucirumab, revealing their potential roles as second-line treatments for hepatocellular carcinoma [13]. Four clinical trials have showcased the effectiveness of three different drugs in treating patients with HCC who are unresponsive to sorafenib. These trials include RESORCE (testing regorafenib), CELESTIAL (testing cabozantinib), and both REACH and REACH-2 (testing ramucirumab) [12]. However, HCC exists within a complex immunological microenvironment, which results in lower remission and survival rates when treated with a single immunotherapy method or immunotherapies alone. Consequently, the emphasis of future development should be on multitarget combination therapy [32,33]. Identifying and developing new targeted agents is of paramount importance for the advancement of targeted therapy.

A significant difference in DRG expression was observed in most tumors, despite a lack of normal comparison in certain cancer types. Of note, both SLC7A11 and SLC3A exhibited marked differential expression across almost all cancers. To further elucidate the potential roles of DRGs in tumorigenesis and the mechanism for their aberrant expression, we conducted a thorough analysis of methylation and mutations across pan-cancer settings. In the pan-cancer mutation landscape, DRGs consistently exhibit a high prevalence of CNV and SNV across diverse tumorigenic contexts. However, our study reveals that SLC7A11 infrequently presents with CNV and SNV, especially in hepatocellular carcinoma. DRGs exhibit a negative correlation with DNA promoter methylation across diverse tumor classifications. Particularly, the methylation status of SLC7A11 consistently correlates with a less favorable prognosis pronounced in LIHC. In conjunction with the established framework of disulfidoptosis, the mentioned results emphasize the pivotal function of SLC7A11 and hint at its underlying mechanism. Grounded on the findings of SLC7A11, we suggest a potential therapeutic strategy that might improve clinical outcomes for oncology patients, involving manipulating the methylation profile and modulating disulfidoptosis-driven tumorigenic activities.

While exploring the relationship between DRGs and the TME, evident variances were found across diverse immune subtypes. The expression of DRGs was inversely correlated with both ImmuneScore and StromalScore and positively correlated with tumor purity, as illustrated by the TME analyses. The correlation analysis between disulfidoptosis and TME, as well as immune infiltration patterns, may unveil a potential scenario where patients with heightened DRG expression might be in an immunologically desert TME state. These findings mentioned above not only elucidate the oncogenic properties of DRGs but also emphasize their prospective significance as therapeutic targets.

Based on the results from the Cox analysis and KM survival studies, subsequent research primarily focused on LIHC. Among our investigations into hepatocellular carcinoma, single-cell sequencing was notably significant. This advanced approach allowed us to dissect the cellular milieu of HCC and unveil the complicated cellular heterogeneity that often remains concealed in bulk RNA sequencing [34].

Through log-normalization and dimensionality reduction, our single-cell landscape revealed distinct subgroups within HCC. Each of these subclusters represented a unique cellular identity, characterized by its own molecular signature, which may potentially be associated with their specific functions within the tumor microenvironment.

The Treg cell populations we focused on, especially those expressing marker genes like *FOXP3*, *CCR8*, *TNFRSF8*, *LAYN*, *TNFRSF9*, *IKZF2*, *RTKN2*, *CTLA4*, *BATF*, and *IL21R*, played a pivotal role in maintaining immune balance [21]. Alternatively, Treg cells have an anti-tumor immune role by reducing the immunity of tumor-associated antigen-specific T cells [35]. Considering both aforementioned points, the expression patterns of these marker genes across different clusters provided not only a snapshot of the T-cell repertoire within HCC but also hinted at the dynamic interplay of immune activation and suppression.

The role of DRGs in this cellular tapestry was another pivotal revelation. Given their known impacts on the survival and death of tumor cells, their differential expression across cellular subpopulations may suggest nuanced roles in determining cellular outcomes. For instance, the distinction of cellular clusters between high and low DRG expression may hint at differing metabolic states, proliferative capacities, and even influences on therapeutic interventions.

Next, we investigated the immune and biological functions of DRGs. Using the GSVA method, we assessed the biological processes associated with the 10 DRGs. The expression of DRGs is primarily positively correlated with several tumoral pathways, DNA replication, and the cell cycle, while it is negatively correlated with tumor metabolism. These findings mentioned that cellular proliferation and metabolism were related to the expression of DRGs, which could potentially act as a prognostic marker for tumor cell progression in HCC. Then, the analysis of TME cell infiltration revealed that the expression of DRGs was negatively correlated with immune-, stromal-, and ESTIMATE scores. Furthermore, it indicated a significant positive correlation between the expression of DRGs and naive CD4 T- and B-cells, Treg, M0 macrophages, resting mast cells, and dendritic cells. Surprisingly, conversely, there was a negative correlation between DRGs and M1 macrophages, M2 macrophages, activated mast cells, and dendritic cells. This observation may provide a potential explanation for the result of the aforementioned immune infiltration analysis. These analyses examining the correlation between disulfidoptosis and TME, as well as immune infiltration patterns, suggest a potential scenario wherein patients exhibiting higher DRG expression may present an immunologically quiescent TME state. These results not only elucidate the oncogenic characteristics of DRGs but also emphasize their prospective significance as therapeutic targets.

HCC, characterized by its distinct molecular pathogenesis and a complex TME, advances rapidly, significantly limiting the efficacy of conventional therapies [36]. Prior studies have revealed the significantly upregulated expression of SLC7A11 in tumor-associated macrophages (TAM). And knocking out SLC7A11 in these macrophages significantly diminished the infiltration of TAMs and hindered the shift to an M2-like phenotype in HCC tissues, which in turn attenuated tumor growth and metastasis [37]. Prior investigations have concentrated on the role of ferroptosis in tumor cells, highlighting how SLC7A11-mediated ferroptosis and phenotypic alterations in TAMs profoundly modify the HCC tumor microenvironment and promote tumor proliferation [38]. Studies have revealed that the silence of SLC7A11 impairs macrophage recruitment and polarization, primarily by inhibiting the secretion of M2 phenotype inducers and cytokines and disrupting the IL-4-driven SOCS3-STAT6-PPAR-γ signaling axis [14]. Furthermore, ferroptosis in macrophages, driven by SLC7A11, significantly increases PD-L1 expression. The therapeutic efficacy is notably enhanced when SLC7A11-specific knockout in macrophages is combined with anti-PD-L1 therapy, surpassing the outcomes of either approach used independently [39]. This suggests that a combination of SLC7A11-targeted therapies and immunotherapies may offer superior therapeutic benefits clinically. Our research aims to delve deeper into SLC7A11-mediated disulfidoptosis, underpinning its viability as a target for HCC treatment.

Finally, utilizing CCK8, colony formation, and Transwell assays, we demonstrated that SL7CA11 acts as a driving factor for proliferation, migration, and invasion in hepatocellular carcinoma cell lines.

Our study presented several inherent limitations. Firstly, the data utilized in this article were derived exclusively from the TCGA database alone, without inclusion from our center or external validation in other public databases or centers. Secondly, further explorations in vivo and in vitro are essential to comprehensively understanding the role of DRGs in HCC. And our conclusions also need further animal experiments and subsequent clinical data to solidify our findings. Additionally, experiments using immune checkpoint inhibitors in immunotherapy and second-line therapy are crucial to confirm the potential of DRGs as viable targets in such treatments.

## 5. Conclusions

In conclusion, through a comprehensive pan-cancer analysis, our research demonstrated the significant prognostic and immunological role of disulfidoptosis across a spectrum of tumors, with particular relevance to LIHC. Notably, our findings highlight the potential of disulfidoptosis-related genes as promising therapeutic targets.

## Figures and Tables

**Figure 1 biomedicines-12-00267-f001:**
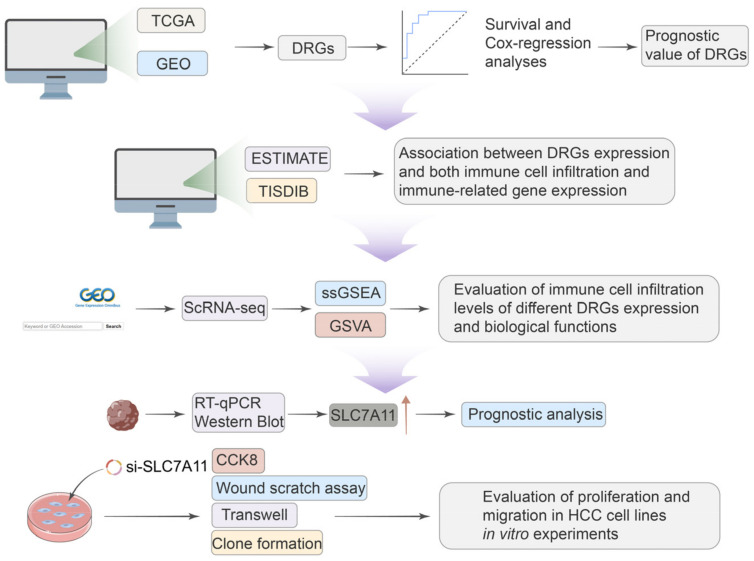
A workflow of the study.

**Figure 2 biomedicines-12-00267-f002:**
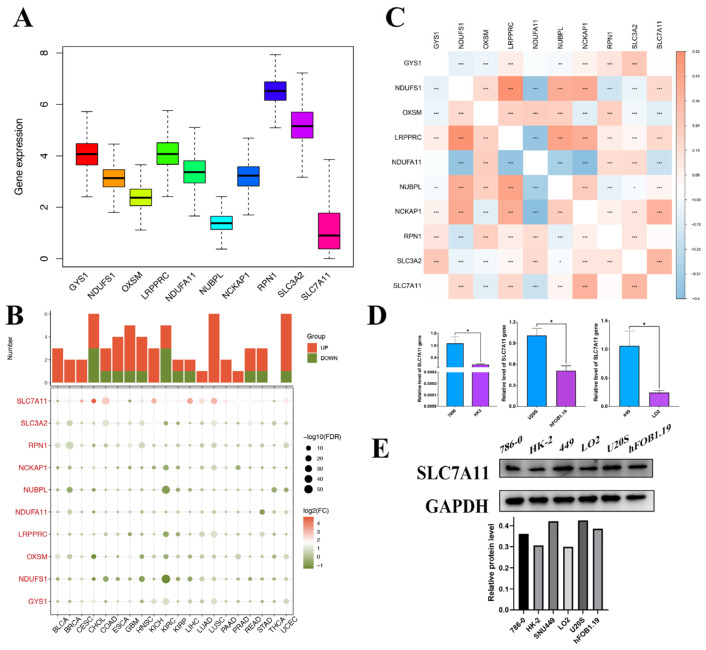
Pan-cancer gene expression profiles of disulfidoptosis-related genes. (**A**) Expression box diagram of DRG expression in pan cancer in TCGA. (**B**) Heatmap showing the expression of each DRG in different tumor and normal tissues. The size of the circles indicates statistical significance (FDR). The expression data consist of log2-transformed fold change (FC) values (log2FC). Color coding indicates the log2(FC) of induction (red) or repression (green). (**C**) Correlation heatmap between each DRG. (**D**) Results from qPCR validation. The relative mRNA expression of SLC7A11 in each tumor cell line is relative to GAPDH in normal cell lines. (* *p* < 0.05) (**E**) WB validation results. The protein expression of SLC7A11 in tumor cell lines and normal cell lines. DRG, disulfidoptosis-associated genes; FDR, false discovery rate; qPCR, quantitative polymerase chain reaction; and WB, Western blot. * *p* < 0.05; ** *p* < 0.01; and *** *p* < 0.001.

**Figure 3 biomedicines-12-00267-f003:**
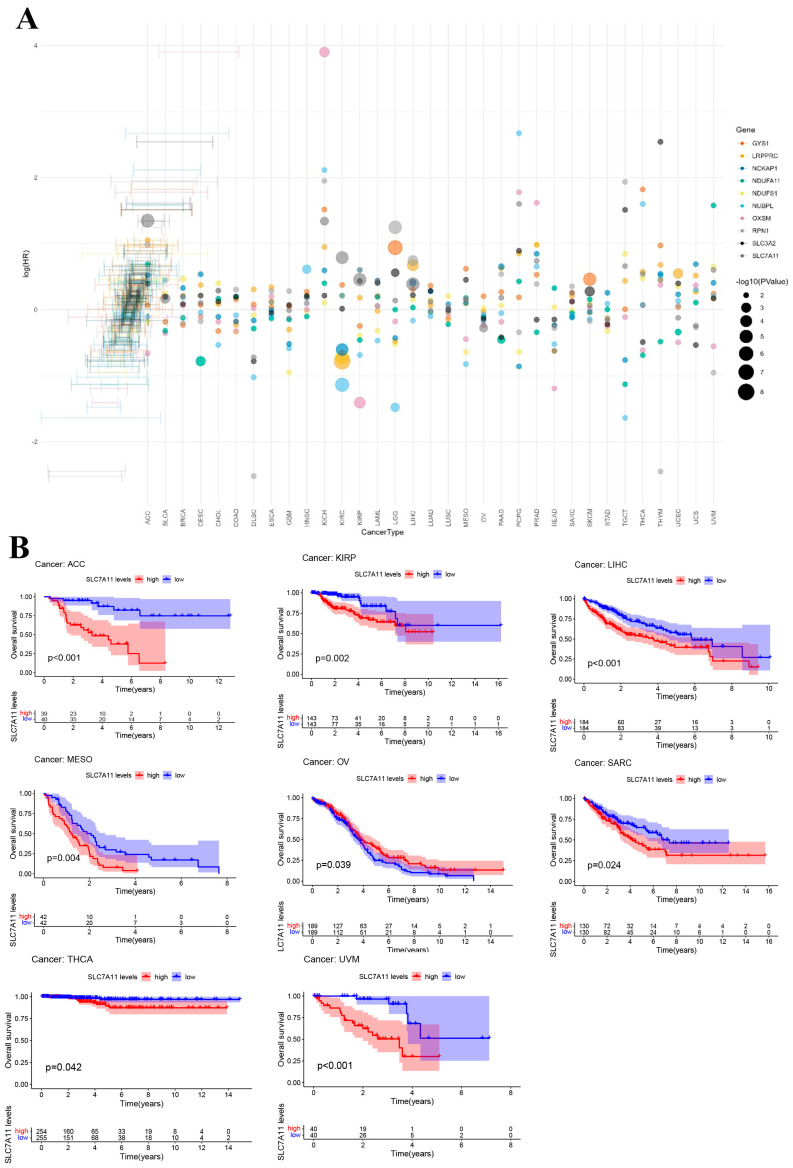
Prognostic value of disulfidoptosis-related genes in pan cancer. (**A**) Forest plot of the survival analysis for the disulfidoptosis-related gene in pan cancer. (**B**) Kaplan–Meier (KM) survival curve of SLC7A11 in ACC, KIRP, LIHC, MESO, OV, SARC, THCA, and UVM.

**Figure 4 biomedicines-12-00267-f004:**
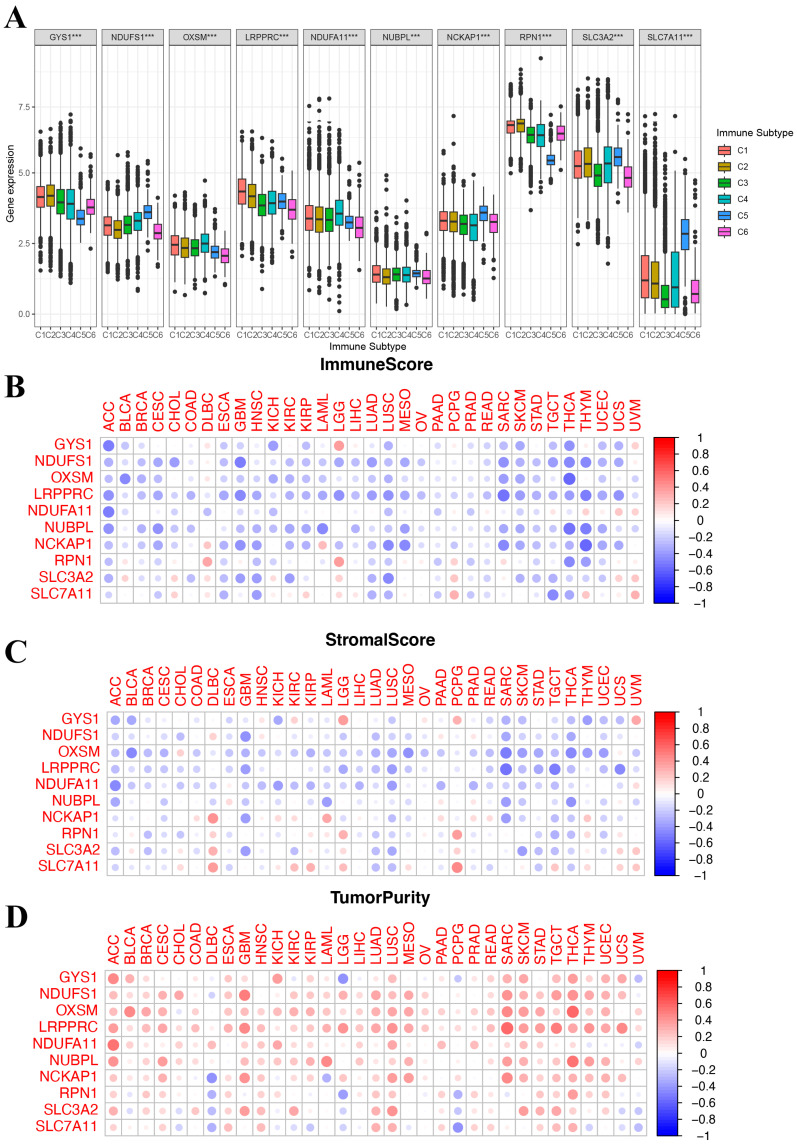
Correlation of disulfidoptosis-related gene expression with the tumor immune microenvironment. (**A**) The relationship between disulfidoptosis-related gene expression and pan-cancer immune subtypes. (**B**–**D**) Heatmap displaying correlations between disulfidoptosis-related gene expression and immune, stromal, and tumor purity scores (from ESTIMATE). Red: high level of immune cell infiltration; blue: low level of immune cell infiltration; the size of the circles indicates statistical significance. *** *p* < 0.001.

**Figure 5 biomedicines-12-00267-f005:**
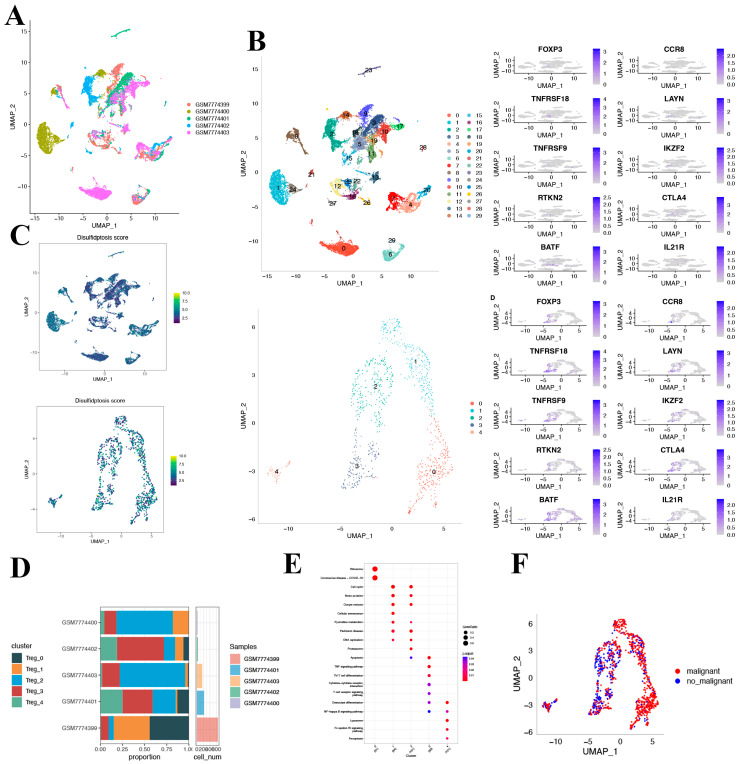
Characterization of Treg cell clusters using scRNA-seq data from HCC patients. (**A**) UMAP visualization illustrating the distribution of 5 samples. (**B**) UMAP representation highlighting the distribution of 30 identified clusters, further distinguishing five Treg cell subsets post-clustering. (**C**) The activity score of disulfidoptosis in each cell. (**D**) A detailed breakdown of subgroups within the cancerous tissue, complemented by the proportion and cell count in adjacent non-cancerous tissue. (**E**) KEGG pathway enrichment analysis corresponding to the five identified Treg cell subsets. (**F**) UMAP visualization depicting the spatial distribution of cells classified as malignant or non-malignant based on predictions from the copykat package.

**Figure 6 biomedicines-12-00267-f006:**
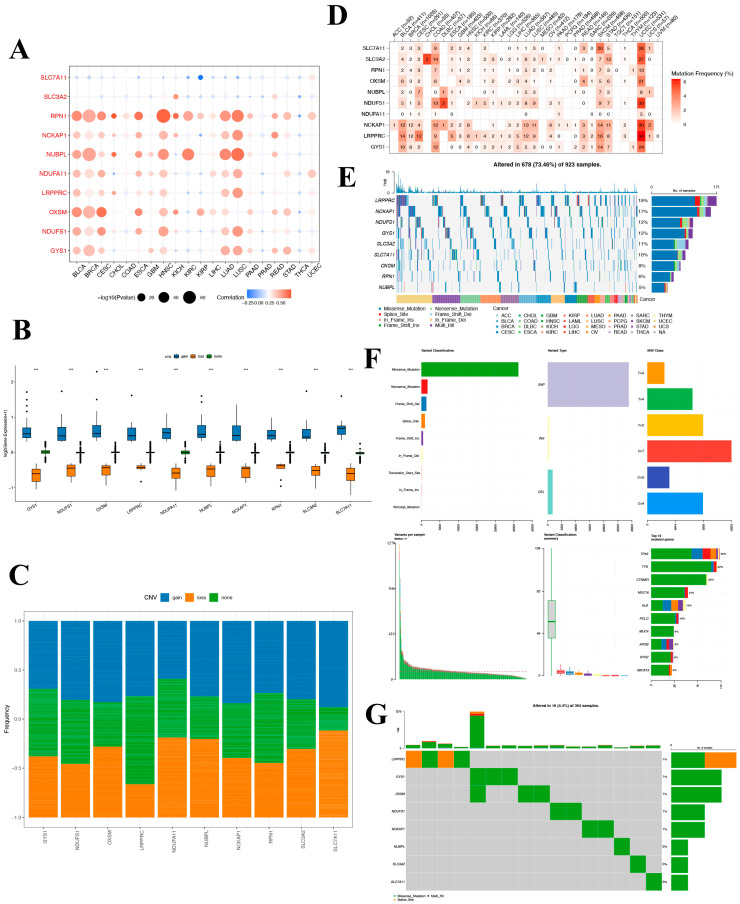
Mutation landscape of disulfidoptosis-related genes in cancers. (**A**) A CNV heat map of the disulfidoptosis-related genes, where red represents increased CNV and blue represents decreased CNV. The size of the circles indicates statistical significance (−log10 *p* value). (**B**) Distribution of three distinct CNV types among the disulfidoptosis-related genes. (**C**) CNV frequency of pyroptosis-regulated genes in TCGA-LIHC samples. (**D**) SNV mutation frequency of disulfidoptosis-related genes in TCGA pan-cancer. (**E**) SNV waterfall plot showing the mutation distribution of disulfidoptosis-related genes and a classification of variant SNV types in TCGA pan-cancer. (**F**,**G**) Somatic mutation landscape and waterfall chart representation for disulfidoptosis-related genes in the TCGA-hepatocellular carcinoma cohort. Out of the 364 HCC patients, 16 exhibited genetic alterations in the 8 disulfidoptosis-related genes, resulting in an alteration frequency of 4.4%. The numbers on the right display the mutation frequency for each respective gene, while individual columns represent distinct patients.

**Figure 7 biomedicines-12-00267-f007:**
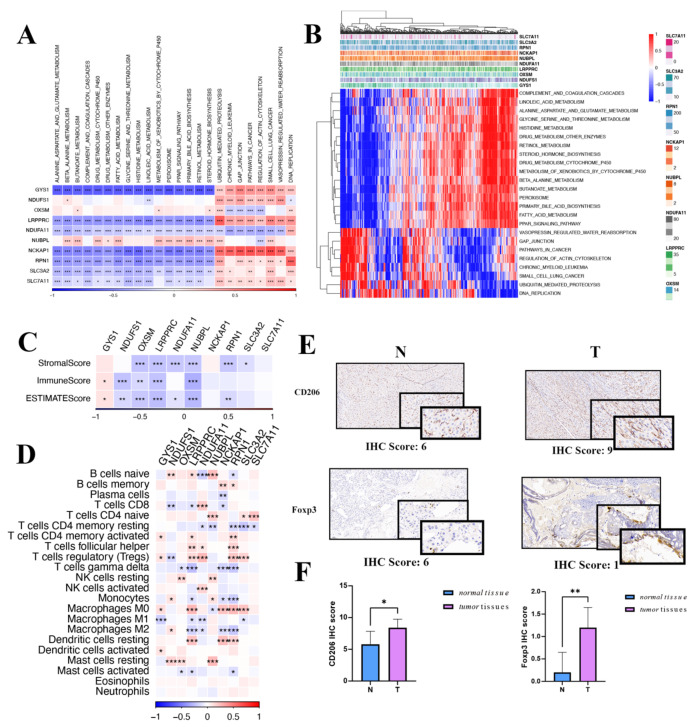
Gene set enrichment analysis and immune landscape of the disulfidoptosis-related genes in HCC. (**A**) A correlation heatmap of disulfidoptosis-related gene expression and pathway activity. (**B**) Heatmap illustrating gene-pathway correlations. (**C**) Associations between disulfidoptosis-related gene expression and immune-, stromal-, and ESTMATE scores are shown. (**D**) Correlation heatmap of disulfidoptosis-related genes with immune cell infiltration. (**E**) IHC scoring in both tumor and normal tissues for macrophages and Treg cells. (**F**) Differential IHC staining scores in tumor versus normal tissues for CD206 and FOXP3. Red indicates activated pathways, while blue denotes inhibited pathways. IHC: immunohistochemistry; Treg: regulatory T cells. * *p* < 0.05, ** *p* < 0.01, and *** *p* < 0.001.

**Figure 8 biomedicines-12-00267-f008:**
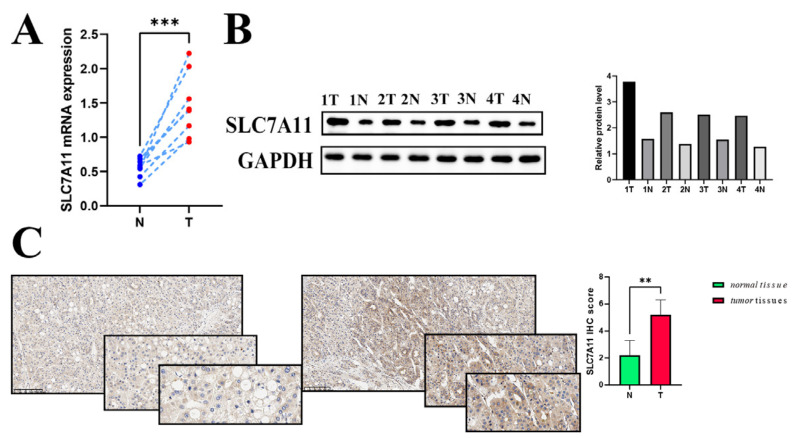
External validation of SLC7A11 expression (**A**,**B**) The mRNA level (relative to GAPDH) and the protein expression of SLC7A11 in hepatocellular carcinoma tissues compared to each corresponding normal adjacent tissue (*p* < 0.05). (**C**) IHC scoring in both hepatocellular carcinoma tissues and normal liver tissues. Differential IHC staining scores in tumor versus normal tissues for SLC7A11. IHC: immunohistochemistry; ** *p* < 0.01, and *** *p* < 0.001.

**Figure 9 biomedicines-12-00267-f009:**
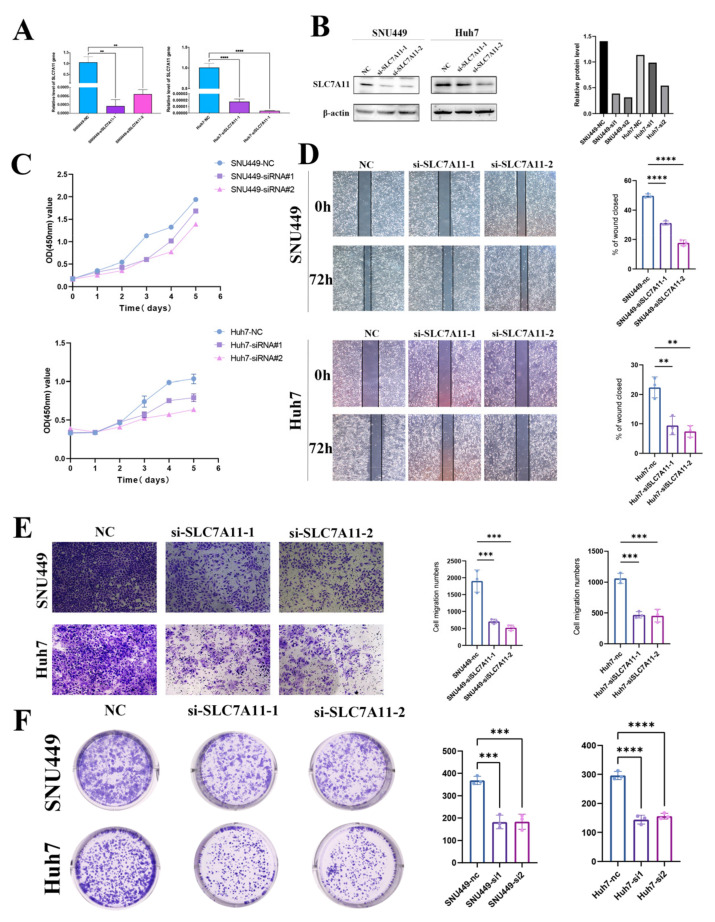
Effect of SLC7A11 silencing in LIHC cell lines. (**A**,**B**) Reduced SLC7A11 mRNA and protein levels in siRNA-SLC7A11-transfected hepatocellular carcinoma cells were verified using real-time PCR (**A**) and Western blotting (**B**). (**C**) The effects of SLC7A11-siRNA on cell proliferation were analyzed with a CCK-8 assay and measured as OD values. (**D**) Representative wound-healing images at 0 and 72 h (magnification: ×100) in 2 hepatocellular carcinoma cell lines. The experiment was repeated three times, independently. The scratch width had decreased significantly in the blank control after 72 h, whereas there was still a wider gap in the siRNA-SLC7A11 cells. (**E**) Transwell migration assay to analyze the migration of hepatocellular carcinoma cells. (**F**) colony formation assays to detect hepatocellular carcinoma cell proliferation. ** *p* < 0.01; *** *p* < 0.001; **** *p* < 0.0001 and ns, not significant.

## Data Availability

Data generated or used in this study are available from the corresponding authors upon reasonable request.

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
