# Peer review of "Pan-Cancer Analysis Reveals Disulfidoptosis-Associated Genes as Promising Immunotherapeutic Targets: Insights Gained from Bulk Omics and Single-Cell Sequencing Validation"

_biomedicines, 2024, doi:10.3390/biomedicines12020267_

Round 1

Reviewer 1 Report

Comments and Suggestions for Authors

The authors conducted a comprehensive analysis of disulfidoptosis-associated genes (DRGs) across 33 cancer types. They analyzed DRG expression levels using several large datasets from TCGA, GEO and ICGC. Survival and Cox regression analysis revealed the prognostic value of DRGs. DRG expression was analyzed in relation to immune cell infiltration and immune-related gene expression using ESTIMATE and TISDIB datasets. Single-cell RNA sequencing data from GEO was used to assess disulfidoptosis activity in various cell types. Biological functions and immune cell infiltration for different DRG expression levels were evaluated using ssGSEA and GSVA. Real-time PCR and Western blot were used to explore SLC7A11 expression in hepatocellular carcinoma tissues and cell lines. Knockdown of SLC7A11 with siRNA in HCC cell lines investigated effects on proliferation and migration. DRG expression, especially SLC7A11, was significantly higher in tumors and associated with poorer outcomes. Except SLC7A11, DRGs had high copy number and single nucleotide variation rates in HCC. DRG expression correlated with tumor stemness indices and inversely with methylation across tumors. TME analysis showed DRG expression negatively correlated with ImmuneScore and StromalScore and positively with tumor purity. Analysis of HCC scRNA-seq data revealed diverse cellular subgroups with a focus on Treg cells, providing insights into immune activation and suppression in the TME. Results were validated experimentally. Knockdown of SLC7A11 suppressed proliferation and migration in HCC cell lines. The pan-cancer analysis demonstrated the prognostic and immunological role of disulfidoptosis across cancers, particularly in LIHC. DRGs were highlighted as potential therapeutic targets.

Here are some potential limitations of the study methods and some suggestions:

  • Scratch assay only measures migration over a short time period (76 hours) and 2D environment, which may not fully capture cell migration ability in vivo. A transwell or spheroid migration assay could provide a more physiologically relevant assessment.

  • Western blot only evaluates bulk protein levels and does not account for cell-to-cell variability that may be captured using techniques like immunofluorescence or spatial proteomics.

  • Reliance on publicly available datasets for survival and bioinformatics analyses removes experimental control and validation could be needed. Replication in independent cohorts would strengthen conclusions.

  • Differential expression and survival analyses do not prove causality - functional experiments are needed to validate genes as drivers of tumorigenesis.

  • scRNA-seq data comes from a single study on HCC, so cell types identified may not apply to other cancer types analyzed. More replication would improve generalizability.

  • Mechanistic insights are limited since only basic proliferation/migration assays are done - more detailed functional assays could provide insight into specific tumor-promoting pathways.

  • Sample sizes for patient tissues (n=4 pairs) are small, so statistical power for qPCR validation is limited. Larger cohorts could affirm results.

  • Limitations of the databases and algorithms used for bioinformatics predictions should be acknowledged.

Suggestions include: conducting 3D/in vivo validation, single-cell validation methods, supporting causality with loss/gain of function models, replicating analyses in independent cohorts/datasets, using larger tissue samples, and discussing limitations of computational predictions.

Comments on the Quality of English Language

The authors conducted a comprehensive analysis of disulfidoptosis-associated genes (DRGs) across 33 cancer types. They analyzed DRG expression levels using several large datasets from TCGA, GEO and ICGC. Survival and Cox regression analysis revealed the prognostic value of DRGs. DRG expression was analyzed in relation to immune cell infiltration and immune-related gene expression using ESTIMATE and TISDIB datasets. Single-cell RNA sequencing data from GEO was used to assess disulfidoptosis activity in various cell types. Biological functions and immune cell infiltration for different DRG expression levels were evaluated using ssGSEA and GSVA. Real-time PCR and Western blot were used to explore SLC7A11 expression in hepatocellular carcinoma tissues and cell lines. Knockdown of SLC7A11 with siRNA in HCC cell lines investigated effects on proliferation and migration. DRG expression, especially SLC7A11, was significantly higher in tumors and associated with poorer outcomes. Except SLC7A11, DRGs had high copy number and single nucleotide variation rates in HCC. DRG expression correlated with tumor stemness indices and inversely with methylation across tumors. TME analysis showed DRG expression negatively correlated with ImmuneScore and StromalScore and positively with tumor purity. Analysis of HCC scRNA-seq data revealed diverse cellular subgroups with a focus on Treg cells, providing insights into immune activation and suppression in the TME. Results were validated experimentally. Knockdown of SLC7A11 suppressed proliferation and migration in HCC cell lines. The pan-cancer analysis demonstrated the prognostic and immunological role of disulfidoptosis across cancers, particularly in LIHC. DRGs were highlighted as potential therapeutic targets.

  • Hepatocellular carcinoma (HCC) has a poor prognosis and high rates of recurrence after first-line therapy. Discussion of second-line treatment options and their efficacy would provide valuable clinical context.

  • A systematic review could synthesize the evidence on various second-line therapies used in practice, such as sorafenib, regorafenib, cabozantinib, immunotherapy, etc. This would help inform treatment decisions.

  • A meta-analysis assessing outcomes like overall survival, progression-free survival, response rates, and toxicities across eligible studies would provide quantitative insights into relative benefits of different second-line options.

  • Relating findings to molecular markers discussed in this study, like SLC7A11 expression levels, could help identify predictive biomarkers for second-line responses.

  • Understanding implications for prognostic stratification and personalized treatment selection based on predictive markers identified.

  • Highlighting gaps in existing evidence and priorities for future research, e.g. sequenced vs. combination therapy approaches.

  • Linking back discussions of tumor microenvironment, immune contexture, and potential immunotherapeutic strategies would appeal to an immuno-oncology readership (bayesian network meta-analysis and second line treatment for hcc are available).

Reviewer 2 Report

Comments and Suggestions for Authors

In this manuscript “Pan-cancer analysis reveals Disulfidoptosis-Associated Genes as Promising Immunotherapeutic Targets: Insights Gained from Bulk Omics and Single-Cell Sequencing Validation”, Xu et al., analyzed the marker genes associated with Disulfidoptosis. The analysis summarized expression profiles, copy number, mutational, and methylation profiles of these genes. Further they did validations using internal data set. Along with bulk analysis, authors have summarized these genes expression in one of the single cell data sets.

The following major and minor points needs to be addressed,

1.     Make the workflow (Figure 1) simpler, as it is very hard to comprehend in the current format without any explanation. It is better to remove all the images and just a simple flow diagram would do.

2.     Line 130 (Reference of former studies for marker genes) are missing.

3.     What is DGEs in line 132, as it has not been introduced previously?

4.     Line 150 where this DGEs come from? Try to use one abbreviation throughout the manuscript in order to avoid confusion.

5.     Line 150 and 151, correction required from promote to promoter

6.     Line 171 spelling correction “orrelation” to correlation

7.     Line 298, did the analysis excluded normal samples from the pan cancer cohort? If yes, please mention it clearly.

8.     Figure 2B, which correlation test was used and what was the p-value threshold, as the correlation values are not significantly higher in majority of the cases?

9.     For Figure 2C, the pattern of SLC7A11 is quite opposite to all the genes across pancan. The color scheming in the heatmap does not map with the supplementary Figure 2, which shows that most of the genes have significantly higher expression in tumor than normal samples. While Figure 2c reflects kind of opposite pattern due to chosen color scheme, which is not consistent with the pattern and does not correlate with Figure 2B as well.

10.  From Line 316 to 319 explanation should be combined with Figure 2C, as it is part of the same analysis.

11.  Supplementary Figure 2 and 3 legends have been interchanged, correction needed.

12.  The visualization of forest plot Figure 3A is not clear as it is hard to see the HR values off the line. No statistical test has been mentioned here to get the differential pattern. If a significance test was used it should be indicated with the forest plot to highlight the significant cases.

13.  Supplementary Figures 5D and E should be excluded as it does not add much towards LIHC which is the major focus for most of the results, as the correlation is hardly detected and is pretty weak.

14.  Add statistical information to Supp. Figures S6 and 7 and based on those redefine the results, as correlation looks pretty weak especially for LIHC for most of cases.

15.  Line 439, spelling correction “edivent” to evident

16.  Single cell analysis does not correlate with the bulk TCGA analysis. The identification of cell types and rational of picking up only Treg is not clear. Further utilizing the module score, Figure 5C highlights some clusters (e.g., Cluster 1) which have higher score but they are not Tregs. What cell type is cluster 1 biologically highlighted in Figure 5B? Which clusters reflect the hepatocytes in single cell data and how they correlate with Treg or other immune related clusters which are not considered here? This analysis needs major revision and a strong connection with the bulk data, otherwise it seems irrelevant here.

17.  From Sections 3.7 onwards should be linked with TCGA data, as this is bulk data with validation. It should be combined and addressed along with the TCGA data, as it is disconnected. All the bulk analysis along with validations should be synchronized together and then the single cell should be revisited.

Comments on the Quality of English Language

Corrections mentioned along with other comments.

Reviewer 3 Report

Comments and Suggestions for Authors

The authors made an effort to show that one of DRGs, SLC7A11 played an important role in malignant potential in HCC cell lines based on disulfidoptosis cell death context. They performed a lot of analysis using commercially available databases, however, the experiments to validate the data were pretty poor and the results which were shown here are deemed to be preliminary.

1.       According to the supplementary figure 1, high SLC7A is associated with cell death. Why did the high expression in cancer cells especially HCC relate to the malignant potential? This point is very complicated, and the authors should discuss it.

2.       The abstract is long and verbose. It should be summarized more concisely.  

3.       Abbreviations such as types of tumors should be clearly written in the main text. It is hard to understand.

4.       Figure 1: This is not the workflow but a showcase of Figure 2-8. They need to show the detail of the methodology here.

5.       Line 78-79: What kind of drugs are related?

6.       The blots (Figure 2E, Figure 8D) were hard to interpret. It did not seem so different between tumor cell lines and normal lines in Fig2E. Also, there is no change in control and siRNA-1 in Huf7 cells in Fig8D. They need to show much clearer bands.

7.       They did not perform any experiments regarding immunotherapy using immune checkpoint inhibitors. They cannot conclude DRG is a promising target for immunotherapy here.

8.       Line 705-707: the format sentence should be excluded.

Round 2

Reviewer 1 Report

Comments and Suggestions for Authors

The authors partially addressed my comments. Raw data should be provided regarding the western blots. Moreover, 

In the discussion, the authors explore the critical roles of cell proliferation and programmed cell death in tumor development, emphasizing the potential of inducing non-programmed necrosis in cancer treatment. They discuss various forms of cell death, including disulfidoptosis, and its relevance to tumor cell death, specifically in hepatocellular carcinoma (HCC). The manuscript underscores the intricate relationship between disulfidoptosis, reactive oxygen species (ROS) accumulation, and energy metabolism, pointing to potential therapeutic insights.

The study highlights differential expression of Death Resistance Genes (DRGs) across various cancers and emphasizes their correlation with DNA replication, cell cycle, and tumor metabolism. The authors delve into the mutation landscape of DRGs and their association with diverse tumorigenic contexts. They propose manipulating the methylation profile and modulating disulfidoptosis as a potential therapeutic strategy, particularly focusing on SLC7A11.

The authors employ single-cell sequencing to explore the cellular milieu of HCC, revealing distinct subgroups and emphasizing the role of Treg cell populations in immune balance. DRGs' expression patterns across different cellular clusters hint at nuanced roles influencing metabolic states and therapeutic interventions.

The discussion extends to the immune and biological functions of DRGs, revealing correlations with tumoral pathways, DNA replication, cell cycle, and TME cell infiltration. The study suggests a potential scenario where higher DRGs expression correlates with an immunologically quiescent TME state, emphasizing DRGs' prospective significance as therapeutic targets.

In the context of HCC, the manuscript discusses the limitations of conventional therapies and the upregulated expression of SLC7A11 in tumor-associated macrophages. The authors propose SLC7A11-mediated disulfidoptosis as a viable target for HCC treatment, highlighting its potential combination with immunotherapies for enhanced therapeutic benefits. Experimental evidence using CCK8, colony formation, and Transwell assays supports SLC7A11 as a driving factor for proliferation, migration, and invasion in HCC cell lines.

Despite presenting valuable insights, the study acknowledges limitations, such as exclusive reliance on TCGA data, the need for external validation, and the importance of further in vivo, in vitro, and clinical studies to confirm DRGs as viable targets, particularly in the context of immune checkpoint inhibitors and second-line treatments for HCC.

Hepatocellular carcinoma, Network meta-analysis and Second-line treatment are crucial (please refer to PMID: 34146196 and expand).

Comments on the Quality of English Language

The authors partially addressed my comments. Raw data should be provided regarding the western blots. Moreover, 

In the discussion, the authors explore the critical roles of cell proliferation and programmed cell death in tumor development, emphasizing the potential of inducing non-programmed necrosis in cancer treatment. They discuss various forms of cell death, including disulfidoptosis, and its relevance to tumor cell death, specifically in hepatocellular carcinoma (HCC). The manuscript underscores the intricate relationship between disulfidoptosis, reactive oxygen species (ROS) accumulation, and energy metabolism, pointing to potential therapeutic insights.

The study highlights differential expression of Death Resistance Genes (DRGs) across various cancers and emphasizes their correlation with DNA replication, cell cycle, and tumor metabolism. The authors delve into the mutation landscape of DRGs and their association with diverse tumorigenic contexts. They propose manipulating the methylation profile and modulating disulfidoptosis as a potential therapeutic strategy, particularly focusing on SLC7A11.

The authors employ single-cell sequencing to explore the cellular milieu of HCC, revealing distinct subgroups and emphasizing the role of Treg cell populations in immune balance. DRGs' expression patterns across different cellular clusters hint at nuanced roles influencing metabolic states and therapeutic interventions.

The discussion extends to the immune and biological functions of DRGs, revealing correlations with tumoral pathways, DNA replication, cell cycle, and TME cell infiltration. The study suggests a potential scenario where higher DRGs expression correlates with an immunologically quiescent TME state, emphasizing DRGs' prospective significance as therapeutic targets.

In the context of HCC, the manuscript discusses the limitations of conventional therapies and the upregulated expression of SLC7A11 in tumor-associated macrophages. The authors propose SLC7A11-mediated disulfidoptosis as a viable target for HCC treatment, highlighting its potential combination with immunotherapies for enhanced therapeutic benefits. Experimental evidence using CCK8, colony formation, and Transwell assays supports SLC7A11 as a driving factor for proliferation, migration, and invasion in HCC cell lines.

Despite presenting valuable insights, the study acknowledges limitations, such as exclusive reliance on TCGA data, the need for external validation, and the importance of further in vivo, in vitro, and clinical studies to confirm DRGs as viable targets, particularly in the context of immune checkpoint inhibitors and second-line treatments for HCC.

Reviewer 2 Report

Comments and Suggestions for Authors

Authors have addressed most of the related concerns.

Author Response

Dear Editors and Reviewers:

Thank you for your letter and for the reviewers’ comments concerning our manuscript entitled “Pan-Cancer Analysis Reveals Disulfidoptosis-Associated Genes as Promising Immunotherapeutic Targets: Insights Gained from Bulk Omics and Single-Cell Sequencing Validation(biomedicines-2683145). We sincerely thank you for acknowledging our efforts in addressing the concerns highlighted in your previous comments. We are grateful for your constructive feedback, which has been instrumental in refining our manuscript. Please be assured that we have taken great care to ensure that all pertinent issues have been thoroughly addressed.

Reviewer 3 Report

Comments and Suggestions for Authors

The manuscript has been successfully revised. I have no further comments.

Author Response

(The authors gave the same response as above.)

Round 3

Reviewer 1 Report

Comments and Suggestions for Authors

The authors have clarified several of the questions I raised in my previous review. Most of the major problems have been addressed by this revision.

Comments on the Quality of English Language

The authors have clarified several of the questions I raised in my previous review. Most of the major problems have been addressed by this revision.

Author Response

Dear Reviewer:

Thank you for your letter and for the reviewers’ comments concerning our manuscript entitled “Pan-Cancer Analysis Reveals Disulfidoptosis-Associated Genes as Promising Immunotherapeutic Targets: Insights Gained from Bulk Omics and Single-Cell Sequencing Validation(biomedicines-2683145). We sincerely thank you for acknowledging our efforts in addressing the concerns highlighted in your previous comments. We are grateful for your constructive feedback, which has been instrumental in refining our manuscript. Please be assured that we have meticulously ensured that all relevant issues are thoroughly addressed and have re-examined the manuscript's content with great care.